# Bayesian Optimization for minimizing CVaR under performance constraints

## Abstract

Optimal portfolio allocation can often be formulated as to a constrained risk problem, where one aims to minimize a risk measure subject to some performance constraints. This paper presents new Bayesian Optimization (BO) algorithms for such constrained minimization problems, seeking to minimize the conditional value-at-risk a computationally intensive risk measure, under a minimum expected return constraint. The proposed algorithms utilize a new acquisition function, which drives sampling towards the optimal region. Additionally, a new two-stage procedure is developed, which significantly reduces the number of evaluations of the expensive-to-evaluate objective function. The proposed algorithm's competitive performance is demonstrated through practical examples.

Reviewer 1:
Q1 Q3: addressed in Section 6

Reviewer 2: None

Reviewer 3:

## 1 Introduction

Portfolio optimization is the process of determining the optimal allocation of certain resources. While it is best known for its applications in finance, it also has important applications in many other areas, such as energy (Fleischhacker et al., 2019), healthcare (Kheybari et al., 2023), supply chain management (Hamdi et al., 2018) and artificial intelligence (Ghosh et al., 2022). Significant research has gone into developing methods to find an optimal portfolio allocation based on certain risk measures, such as value-at-risk (VaR) or conditional value-at-risk (CVaR). A typical formulation seeks to minimize such a risk measure, subject to a minimum expected return requirement, or constraint.

When the objective and constraint functions are assumed to be linear and accessible (that is, not black-box), they can easily be solved using classic Linear Programming methods, as demonstrated in Rockafellar and Uryasev (2000) and Krokhmal et al. (2002). Furthermore, when the functions are non-linear but still accessible, one can use alternate traditional optimization methods (see e.g., Gaivoronski and Pflug (2005); El Ghaoui et al. (2003); Alexander et al. (2006)). The assumptions of linearity, differentiability, or accessible objective and constraint functions underlie many traditional portfolio optimization algorithms. In many practical settings, however, the objective (i.e., the risk measure) and/or constraint functions are non-linear, noisy and expensive black-boxes, making these approaches infeasible.

Recently, considerable attention has been devoted to developing methods based on Bayesian Optimization (BO) (Brochu et al., 2010; Frazier, 2018; Garnett, 2023) for minimizing risk measures, primarily due to their ability to deal with noisy, expensive and black-box functions. In this regard, Cakmak et al. (2020) proposes a BO algorithm for the unconstrained optimization of VaR and CVaR of black-box expensive-to-evaluate functions, with randomness induced by an environmental vector. Instead of modelling the objective function directly, the paper proposes modelling the underlying function $f$ as a Gaussian Process (GP) and applying a BO method which jointly selects both a portfolio weight $\mathbf{w}$ and realisation of the environmental

variable $\mathbf{Z}$, using a one-step look-ahead approach based on the knowledge gradient acquisition function (**?**). Furthermore, Nguyen et al. propose two alternate BO approaches for the optimization of VaR (Nguyen et al., 2021b) and CVaR (Nguyen et al., 2021a), which provide certain desirable computational and theoretical properties. More recent work includes Daulton et al. (2022a), which aims to tackle multivariate value-at-risk problems, that is, where the optimization problem has multiple objectives with input noise, and Picheny et al. (2022), which studies the situation that the distribution of $\mathbf{Z}$ is unavailable.

The aforementioned methods are designed for the unconstrained minimization of risk measures. To date, no BO algorithm has been designed specifically for constrained portfolio optimization problems. For general constrained optimization problems, a popular class of BO algorithms incorporate the constraints into the acquisition function design (see, i.e., Gramacy and Lee (2011); Gardner et al. (2014); Gelbart et al. (2014)). More recent advances include Lam and Willcox (2017), Letham et al. (2019) and Eriksson and Poloczek (2021), among others. Whilst these methods are effective, they often require frequent evaluation of the risk measure functions, which is unsuitable for complex allocation problems - as detailed shortly.

The main purpose of this work is to design a constrained BO algorithm specifically for the portfolio allocation problem. In this paper, we introduce new BO methods building on Gardner et al. (2014) and Gelbart et al. (2014), designed to take advantage of two key properties which hold in portfolio allocation problems: 1) the expected return constraint functions are much cheaper to evaluate than the objective function, i.e., the risk measures; 2) the expected return constraints are typically *active* – namely, the optimal solution lies on the boundary of the feasible region defined by the constraints.

Firstly, this paper introduces a two-stage BO adaptation, which reduces the number of full-function evaluations needed to find an optimal solution, significantly reducing the computational cost of the algorithm. Only samples that meet certain criteria are fully evaluated in the second stage; this differs from cascade-based BO approaches (Kusakawa et al., 2022) where all samples in the first stage are used in the second, regardless of their feasibility or promise. Secondly, this work proposes a new acquisition function that encourages more sampling in the near-optimal region, improving the algorithm's performance. The paper also details how the proposed methods can be adapted for batch implementation to take advantage of parallel computing.

As the numerical examples demonstrate, the proposed BO algorithms are highly effective for solving constrained portfolio allocation problems, outperforming existing approaches with a lower computational cost and faster convergence. These improvements are achieved by combining a new acquisition function, a two-stage procedure and the potential for parallel batch implementation.

## 2 Optimal Portfolio Allocation

Suppose an investor seeks to find an optimal allocation to $N$-assets. For the target portfolio, we define an $N$-dimensional vector $\mathbf{w} = (w_1, ..., w_N)$ to represent the capital allocation or *portfolio weights*. Each component $w_i$ corresponds to the fraction of the total capital allocated to the $i^{th}$ asset. The vector $\mathbf{w}$ is defined within the constraints of a feasible set $\mathbb{W} = \{\mathbf{w} \in \mathbb{R}^N | w_i \geq 0, \sum_{i=1}^{N} w_i \leq 1\}$, to ensure that the sum of all weights does not exceed the total available capital, which, without loss of generality, is taken to be 1.

To account for the uncertainty in future asset returns, we introduce random variable $\mathbf{Z}$ representing the random environmental factors that can affect the future return. The return function $f(\mathbf{w}, \mathbf{z})$ represents the forecasted portfolio return for an allocation $\mathbf{w}$ and realisation $\mathbf{z}$ from $\mathbf{Z}$. For clarity: $f(\mathbf{0}, \mathbf{z}) = 0$, as no capital invested means no returns; $f(\mathbf{w}, \mathbf{z}) < 0$ if the portfolio is forecasted to lose money and $f(\mathbf{w}, \mathbf{z}) > 0$ if the portfolio is forecasted to gain money. As an example, $f(\mathbf{w}, \mathbf{z}) = 0.1$ means a forecasted gain of 10%; $f(\mathbf{w}, \mathbf{z}) = -0.2$ means a forecasted loss of 20%.

### 2.1 Risk Measures

VaR is defined as the threshold value $\omega$ such that the probability of the loss exceeding $\omega$ is at most $(1 - \alpha)$. Formally, for a return function $f$, set of portfolio weights $\mathbf{w}$ and VaR threshold $\alpha$, we define VaR as:

$$\text{VaR}_\alpha[f(\mathbf{w}, \mathbf{Z})] = \inf\{\omega : \mathbb{P}(f(\mathbf{w}, \mathbf{Z}) \leq -\omega) \leq 1 - \alpha\}$$

For conciseness, we use the notation $v_f(\mathbf{w}; \alpha)$ to denote $\text{VaR}_\alpha[f(\mathbf{w}, \mathbf{Z})]$ for the remainder of this work. Typical values for $\alpha$ include 99.9% and 97.5%.

CVaR at a specified risk level $\alpha \in (0, 1)$ is the expected loss, assuming that the loss is worse than the VaR threshold. It represents the average of the worst-case losses. Formally, CVaR is defined as:

$$\text{CVaR}_\alpha[f(\mathbf{w}, \mathbf{Z})] = -\mathbb{E}[f(\mathbf{w}, \mathbf{Z})|f(\mathbf{w}, \mathbf{Z}) \leq -v_f(\mathbf{w}; \alpha)]$$

Artzner et al. (1999) establishes key desirable properties for risk measures. Of those, CVaR meets many, including subadditivity, translation invariance, positive homogeneity, and monotonicity[1]. In contrast, VaR often exhibits multiple local extrema and unpredictable behaviour as a function of portfolio positions, limiting its usefulness in portfolio optimization problems, limitations which do not apply to CVaR (Mausser and Rosen, 1999; McKay and Keefer, 1996).

Within portfolio optimization, the chosen risk measure must be able to handle and account for uncertainty induced by the environmental vector $\mathbf{Z}$. Embrechts et al. (2022) establishes a framework for considering the effect of uncertainty around an $\alpha$-quantile level, concluding that (unlike VaR) CVaR remains stable and robust in simulation-based optimization methods with uncertainty. Within this study, we choose CVaR as the objective function risk measure.

## 2.2 Problem Set-up

Let us consider the expected return for a selected portfolio, defined under a set of portfolio weights $\mathbf{w}$ as the expectation over all possible forecasted returns, that is, $\mathbb{E}_\mathbf{Z}[f(\mathbf{w}, \mathbf{Z})]$. An underlying principle in investing is that to compensate investors for taking on greater risk, a riskier asset should generate higher returns. A robust risk measure should reflect this feature. Research (see, e.g., Bali and Cakici (2004); Iqbal et al. (2010)) has shown a positive relationship exists between CVaR and expected returns. A higher expected return can only be obtained by increasing risk exposure; conversely, if an investor wishes to reduce the CVaR of their portfolio, the expected return will reduce too.

A key property of CVaR is that it is monotonic with respect to stochastic dominance of order 1 and order 2 (Pflug, 2000). In financial risk management, this property implies that when comparing two investment options, if one demonstrates a lower risk (as indicated by CVaR) while providing an equal or higher expected return, it is universally more favourable regarding the risk-return balance. These features enable us to use CVaR as a reliable and robust risk measure within portfolio optimization.

For a chosen expected return requirement, stochastic dominance ensures that an investor can identify an optimal portfolio that meets or exceeds this return requirement with the lowest possible risk, as measured by CVaR. Therefore, in this context, an optimal portfolio provides the desired expected return with the lowest possible CVaR. This relationship lays the groundwork for our problem set-up.

For ease of notation, we define the objective function as $g(\mathbf{w})$ and the constraint function as $R(\mathbf{w})$. For a minimum expected return threshold value $r^{\min}$, the complete constrained portfolio optimization problem is as follows.

$$\min_\mathbf{w} g(\mathbf{w}) := \text{CVaR}_\alpha[f(\mathbf{w}, \mathbf{Z})] \tag{1a}$$

$$\text{s.t.} \quad R(\mathbf{w}) := \mathbb{E}_\mathbf{Z}[f(\mathbf{w}, \mathbf{z})] \geq r^{\min} \tag{1b}$$

$$0 \leq w_i \leq 1, \, i = 1, ..., N, \quad \sum_{i=1}^{N} w_i \leq 1. \tag{1c}$$

Within this problem set up, we seek to minimize the CVaR (denoted as $g(\mathbf{w})$) constrained by a minimum expected return threshold ($r^{min}$), defined by the constraint function $R(\mathbf{w})$.

---

[1]Subadditivity ensures that the risk of a combined portfolio does not exceed the sum of the individual risks, implying diversification benefits. Translation invariance guarantees that adding a risk-free asset to a portfolio does not alter the risk measure. Positive homogeneity dictates that scaling a portfolio by a positive factor scales the risk measure by the same factor, ensuring proportionality. Monotonicity ensures that a portfolio with consistently higher returns under all scenarios is considered less risky.

# 3 Bayesian Optimization

Bayesian Optimization - introduced in Močkus (1975) - is a powerful method for solving global optimization problems. The method applies to scenarios where the objective function does not have a closed-form expression, but noisy evaluations can be obtained at sampled points. In this section, we present an adaptation to the BO methods developed in Gardner et al. (2014) and Gelbart et al. (2014), to handle the uncertainty caused by an environmental vector $\mathbf{Z}$.

## 3.1 Unconstrained Bayesian Optimization

BO is a probabilistic framework for optimizing black-box functions based on the GP model. In the unconstrained setting, BO sequentially evaluates an objective function at selected points, from which a GP model of the objective function is constructed. The design point(s) are selected by maximizing an acquisition function, which quantifies a desired trade-off between the exploration and exploitation of the GP model. Commonly used acquisition functions include expected improvement, probability of improvement and upper confidence bounds. The standard BO procedure for the unconstrained global minimization of a function $g(\mathbf{w})$ is given in Alg. 1. We note that, in practice it is often impossible to evaluate the objective function $g(\mathbf{w})$ directly, and instead one can obtain a noisy estimate of it which is denoted as $\hat{g}$ in Alg. 1.

---

**Algorithm 1** Bayesian Optimization

---

**Require:** blackbox model $\pi_g(y|\mathbf{w})$, acquisition function $a(\mathbf{w}, \tilde{g})$
**Ensure:** a local minimizer of $g(\mathbf{w})$
  initialize the training data set $D_0$ using an initial design
  let $t = 0$;
  **while** stopping criteria not met **do**
    let $t = t + 1$;
    construct a GP model $\tilde{g}_{t-1}$ using $D_{t-1}$;
    let $\mathbf{w}_t = \arg\max_{\mathbf{w}} a(\mathbf{w}, \tilde{g}_{t-1})$;
    compute an estimate of $g(\mathbf{w}_t)$ denoted by $\hat{g}_t$;
    let $D_t = D_{t-1} \cup \{\mathbf{w}_t, \hat{g}_t\}$;
  **end while**

---

## 3.2 Bayesian Optimization with Constraints

This section presents the BO algorithm for optimization problems with inequality constraints, largely following Gardner et al. (2014) and Gelbart et al. (2014). Suppose that we have the following constrained optimization problem:

$$\min_{\mathbf{w}} \; g(\mathbf{w}) \text{ s.t. } c_k(\mathbf{w}) \leq 0, k = 1, ..., K. \tag{2}$$

To solve Eq. 2 with BO, all the constraint functions $c_k(\mathbf{w})$ need to be modelled as GPs. Namely, the GP model for the $k$-th constraint $c_k(x)$ is obtained from the constraint training set $C^k = \{(\mathbf{w}_1, c_k(\mathbf{w}_1)), ..., (\mathbf{w}_m, c_k(\mathbf{w}_m))\}$, where the constraint functions are evaluated at each design point. Therefore, when selecting the design points, both the objective and constraints need to be considered, which is accomplished by incorporating the constraints into the acquisition function.

Gardner et al. (2014) propose modifying the *Expected Improvement* (EI) acquisition function. Let $\mathbf{w}^+$ be the current best-evaluated point, that is, $g(\mathbf{w}^+)$ is the smallest in the current training set. We define the improvement as

$$I(x) = \max\{0, g(\mathbf{w}^+) - \tilde{g}(\mathbf{w})\}, \tag{3}$$

where $\tilde{g}(\mathbf{w})$ is the GP model constructed with the current objective training set $D$. The EI acquisition function is defined as

$$EI(\mathbf{w}) = \mathbb{E}[I(\mathbf{w})|D],$$

where the expectation is taken over the posterior of $\tilde{g}(\mathbf{w})$. We further adapt this acquisition function to

account for the constraints. For $k = 1, ..., K$, let $\tilde{c}_k(\mathbf{w})$ be the GP model for the constraint function $c_k(\mathbf{w})$, conditional on the training set $C^k$, and let

$$\text{PF}(\mathbf{w}) = \mathbb{P}(\tilde{c}_1(\mathbf{w}) \leq 0, \tilde{c}_2(\mathbf{w}) \leq 0, ..., \tilde{c}_K(\mathbf{w}) \leq 0), \tag{4}$$

which is the probability that a candidate point $x$ satisfies all the constraints. In our present problem, we only need to consider the case where the constraints are conditionally independent given $x$, as such, we have:

$$\text{PF}(\mathbf{w}) = \prod_{k=1}^{K} \mathbb{P}(\tilde{c}_k(\mathbf{w}) \leq 0). \tag{5}$$

Finally, we define the new acquisition function to be

$$a_{\text{CW-EI}}(\mathbf{w}) = \text{EI}(\mathbf{w})\text{PF}(\mathbf{w}), \tag{6}$$

which is referred to as the constraint-weighted expected improvement (CW-EI) acquisition function in Gardner et al. (2014). The constrained BO algorithm proceeds largely the same as the unconstrained version (Alg. 1), except the following two main differences: (1) the constrained acquisition function in Eq. 6 is used to select the new design points; (2) for each design point, both the objective and constraint functions are evaluated. We hereafter refer to this constrained BO method as CW-EI BO.

Finally, we note that in a class of BO approaches (Fröhlich et al., 2020; Cakmak et al., 2020; Daulton et al., 2022b), the underlying function $f(\mathbf{w}, \mathbf{Z})$ is modelled as a single GP (with $\mathbf{w}$ being the input) for a fixed $\mathbf{Z}$ during the optimization procedure and then $\mathbf{Z}$ is only random at implementation time. Whilst this may be appropriate for many unconstrained problems, it is not for portfolio allocation problems. As explained later, we intend to deal with the CVaR function and the expected return constraint separately, and thus we cannot use this single GP model framework.

## 4 BO for Portfolio Optimization

It is possible to directly apply the existing CW-EI BO algorithm to the portfolio optimization problem. This is achieved through modelling the CVaR objective and expected return constraint functions as separate GPs. To be specific, the CW-EI acquisition function is utilized to propose new portfolio weights $\mathbf{w}$, from which a standard Monte Carlo (MC) simulation obtains the distribution $f(\mathbf{w}, \mathbf{Z})$, to obtain the expected return and CVaR for $\mathbf{w}$ - further detailed in the Appendix. The CVaR and expected return values for the proposed weights $\mathbf{w}$ are then used to update the objective and constraint GPs, respectively. Therefore, in the standard CW-EI BO procedure, a full evaluation of the objective and constraint functions must be performed for each proposed portfolio weight to update the respective GPs.

As shown in Appendix A, it is possible to obtain an accurate estimate of the expected return with a relatively low MC sample size, while a large number of MC samples is required to obtain an accurate estimate of the CVaR (from the distribution $f(\mathbf{w}, \mathbf{Z})$). As such, the computational cost of calculating CVaR, i.e., the objective function, is significantly higher than the expected return constraint. For example, in the numerical experiments provided in Section 5.2, the cost for evaluating the expected return is around 1% of that for evaluating CVaR[2]. Therefore, the computational efficiency of BO algorithms can be enhanced by reducing the number of CVaR evaluations.

### 4.1 Activeness of the Constraint

This section formalizes several assumptions related to the portfolio optimization problem. These assumptions and the subsequent Theorem, allow us to develop a new BO algorithm procedure to take advantage of the computational efficiency gained by reducing the number of objective CVaR evaluations.

Before presenting the assumptions, we clarify our notation. It is important to note that $f(\mathbf{w}, \mathbf{z}) \leq 0$ indicates losses, whereas VaR and CVaR are statements about the losses, so $v_f(\mathbf{w}; \alpha) \geq 0$ and $\text{CVaR}_\alpha[f(\mathbf{w}, \mathbf{Z})] \geq 0$ represent negative returns, or losses.

---

[2]The exact saving depends on the choice of $\alpha$ threshold, where the cost saving increases as $\alpha$ decreases.

Now, let us introduce and prove several assumptions concerning the return function $f(\mathbf{w}, \mathbf{z})$ and the distribution of $\mathbf{Z}$ - which are critical to our proposed method.

**Assumptions 1-4.** *1. $f(\mathbf{w}, \mathbf{z})$ is a continuous function of $\mathbf{w}$ for any fixed $\mathbf{z}$; 2. $f(\mathbf{0}, \mathbf{z}) \equiv 0$; 3. for a given $\mathbf{w} \in \mathbb{W}$ and any fixed $\mathbf{z}$, if $f(\mathbf{w}, \mathbf{z}) \leq 0$, $f(\rho\mathbf{w}, \mathbf{z})$ is a decreasing function of $\rho \in [0, 1]$; 4. there exists $\alpha \in (0, 1)$ such that $v_f(\mathbf{w}; \alpha) \geq 0 \ \forall \ \mathbf{w} \in \mathbb{W}$.*

- Assumption 1 ensures that small changes in portfolio allocation do not lead to abrupt or unpredictable changes in outcomes; a reasonable expectation in most financial models.

- Assumption 2 is straightforward; an absence of investment will result in a neutral (zero) financial return.

- Assumption 3 implies that, if a chosen portfolio allocation results in a loss for a certain scenario, this loss does not increase if the total capital is proportionally reduced[3]; this reflects the intuitive notion that if investing a certain amount leads to a loss, investing less should not lead to a greater loss.

- Assumption 4 implies that there always exists a choice of $\alpha \in (0, 1)$ so that no matter the allocation $\mathbf{w} \in \mathbb{W}$, $v_f(\mathbf{w}; \alpha)$ is positive, i.e., a loss. In simpler terms, no matter the allocation, there always exists some level of risk (represented by $\alpha$), which can be chosen to ensure there is always some risk of loss (as indicated by VaR). This is important, as it allows us - through the appropriate choice of $\alpha$ - to just consider the loss scenarios when evaluating the associated CVaR.

From this, we obtain the following theorem:

**Theorem 1.** *If function $f(\mathbf{w}, \mathbf{Z})$ and distribution $p_{\mathbf{z}}(\cdot)$ satisify assumptions 1-4, $\alpha$ is chosen such that $v_f(\mathbf{w}, \alpha) \geq 0 \ \forall \ w \in \mathbb{W}$, and solutions to the constrained optimization problem 1 exist, then there must exist a solution to problem 1, denoted as $\mathbf{w}^*$, such that $R(\mathbf{w}^*) = r^{min}$.*

*Proof.* First, assume that $\mathbf{w}'$ is a solution to the constrained optimization problem 1. It follows directly that $R(\mathbf{w}') \geq r^{\min}$. Obviously if $R(\mathbf{w}') = r^{\min}$, the theorem holds.

Now consider the case that $R(\mathbf{w}') > r^{\min}$, i.e., it does not lie on the boundary of the feasible region. From assumption 1, $R(\mathbf{w})$ is a continuous function of $\mathbf{w}$ in $\mathbb{W}$. Next define a function

$$h(\rho) = R(\rho\mathbf{w}')$$

for $\rho \in [0, 1]$. As $R(\mathbf{w})$ is a continuous function in $\mathbb{W}$, $h(\rho)$ is a continuous function too.

From assumption 2, we know that $h(0) = 0$, and therefore,

$$h(0) = 0 < r^{\min} < h(1) = R(\mathbf{w}')$$

According to the intermediate value theorem on continuous functions, there exists some $\rho^* \in (0, 1)$ such that $h(\rho^*) = R(\rho^*\mathbf{w}') = r^{\min}$. Let $w^* = \rho^*\mathbf{w}'$ denote this point, which lies on the constraint boundary - we wish to compare $F(\mathbf{w}^*)$ and $F(\mathbf{w}')$, i.e., the CVaR values at these two points for a fixed $\alpha$.

From the Theorem's assumption, we have $v_f(\mathbf{w}', \alpha) \geq 0$ and $v_f(\mathbf{w}^*, \alpha) \geq 0$. From assumption 3, we know that for any $\mathbf{z}$, if $f(\mathbf{w}', \mathbf{z}) \leq 0$, then $f(\mathbf{w}', \mathbf{z}) \leq f(\mathbf{w}^*, \mathbf{z}) \leq 0$.

It follows that for any $\mathbf{z} \in \{\mathbf{z} | f(\mathbf{w}', \mathbf{z}) \leq -v_f(\mathbf{w}', \alpha)\}$, we have

$$f(\mathbf{w}', \mathbf{z}) \leq f(\mathbf{w}^*, \mathbf{z}) \leq -v_f(\mathbf{w}^*, \alpha) \leq 0.$$

---

[3]For clarity, as $\rho$ goes from 0 to 1, $f$ goes from $f(0, \mathbf{z}) \equiv 0$ to $f(\mathbf{w}, \mathbf{z})$. As $f(\mathbf{w}, \mathbf{z}) \leq 0$, the function value $f(\rho\mathbf{w}, \mathbf{z})$ gets more negative, so $f$ is a decreasing function w.r.t. $\rho \in [0, 1]$.

As such, we can derive $v_f(\mathbf{w}^*, \alpha) \leq v_f(\mathbf{w}', \alpha)$, and obtain,

$$
\begin{aligned}
\mathrm{CVaR}_\alpha[f(\mathbf{w}^*, \mathbf{Z})] &= -\mathbb{E}[f(\mathbf{w}^*, \mathbf{Z})|f(\mathbf{w}^*, \mathbf{Z}) \leq -v_f(\mathbf{w}^*; \alpha)] \\
&\leq -\mathbb{E}[f(\mathbf{w}^*, \mathbf{Z})|f(\mathbf{w}^*, \mathbf{Z}) \leq -v_f(\mathbf{w}'; \alpha)] \\
&\leq -\mathbb{E}[f(\mathbf{w}', \mathbf{Z})|f(\mathbf{w}', \mathbf{Z}) \leq -v_f(\mathbf{w}'; \alpha)] \\
&= \mathrm{CVaR}_\alpha[f(\mathbf{w}', \mathbf{Z})].
\end{aligned}
$$

Therefore, $\mathbf{w}^*$ is also a minimal solution w.r.t. the objective function and $R(\mathbf{w}^*) = r^{\min}$. The proof is thus complete. $\qquad\square$

Simply put, Theorem 1 states that under some reasonable assumptions, the constraint 1b is active for at least one solution. The result is rather intuitive, as it infers that a higher expected return can only be obtained by increasing risk exposure and, as such, the CVaR. The optimal solution to our problem will likely arise from an active constraint, where the minimum expected return requirement limits our ability to reduce the CVaR further. This aligns with our earlier analysis of CVaR's properties of stochastic dominance and its relationship to returns. These observations provide a useful heuristic and motivate us to drive sampling towards the active region. We shall also note that, while the assumptions are sensible from the practical point of view, some of them such as those on $\alpha$ are are rather difficult to verify in advance. Further comments on the matter are provided in the conclusions.

## 4.2 Two-Stage Point Selection

Intuitively, Theorem 1 suggests that a solution to problem 1 can be found close to the boundary of the constraint. Therefore, based on the expected return value for a proposed portfolio weight $\mathbf{w}$, evaluating the CVaR objective function is unnecessary under the following two situations.

Firstly, if the expected return is lower than the minimum constraint threshold, the proposed design point is not feasible, so the CVaR function does not need to be evaluated. Secondly, if the expected return is too high (i.e., not approximately active), the corresponding CVaR is likely far from optimal, so the objective does not need to be evaluated.

We introduce a maximum expected return parameter, denoted by $r^{\max}$, set on the basis that those points with expected returns higher than this parameter value are highly unlikely to be optimal for our objective. Based on these observations, we propose a two-stage point selection procedure. The first stage selects a design point based on the acquisition function. In the second stage, the expected return is calculated. If the expected return satisfies the requirement that

$$
r^{\min} \leq R(\mathbf{w}) = \mathbb{E}_{\mathbf{Z}}[f(\mathbf{w}, \mathbf{Z})] \leq r^{\max}, \tag{7}
$$

the more expensive evaluation of our objective function is completed to determine the CVaR value. Then, the GPs for both the constraint and objective functions are updated. If Eq. 7 is not satisfied, the proposed point is rejected, the objective function is not evaluated, and only the GP for the constraint is updated to ensure this point is not re-proposed.

This two-stage (2S) adaptation has the advantage of only fully evaluating those feasible and (approximately) active points. As such, it reduces the number of evaluations of the expensive-to-evaluate CVaR objective. The algorithm obtains two training sets, one for the CVaR objective and one for the expected return, with the former being a subset of the latter.

## 4.3 New Acquisition Function

With the two-stage selection procedure, many more evaluations of the expected return constraint will be completed than the CVaR objective; as such, the GP for the constraint will be more accurate than that of the objective function. As a result, the CW-EI acquisition function will be effective at proposing feasible points due to the quality of the constraint GP but may be poor at proposing points with low CVaR due to

the lower quality of the objective GP. To address this issue, we propose a new acquisition function based on the active constraint assumption.

Namely, as the CW-EI acquisition function only accounts for the feasibility of the constraint, it should be adapted to incorporate the activeness as well. Let $\widetilde{R}(\mathbf{w})$ be a GP model of the expected return $R(\mathbf{w})$, define

$$\text{PF}(\mathbf{w}) = \mathbb{P}(r^{\min} \leq \widetilde{R}(\mathbf{w}) \leq r^{\max}) \tag{8}$$

as the probability that a chosen weight $\mathbf{w}$ is feasible and approximately active. Therefore, for $\mathbf{w}$,

$$\begin{aligned} \text{PF}(\mathbf{w}) &= \text{PF}_{\min}(\mathbf{w}) \times \text{PF}_{\max}(\mathbf{w}) \\ \text{PF}_{\min}(\mathbf{w}) &= \mathbb{P}(\widetilde{R}(\mathbf{w}) \geq r^{\min}) \\ \text{PF}_{\max}(\mathbf{w}) &= \mathbb{P}(\widetilde{R}(\mathbf{w}) \leq r^{\max}) \end{aligned} \tag{9}$$

Combining Eq. 9 with the Expected Improvement, obtains:

$$a_{\text{ACW-EI}}(\mathbf{w}) = \text{EI}(\mathbf{w})\text{PF}_{\min}(\mathbf{w})\text{PF}_{\max}(\mathbf{w}), \tag{10}$$

which is hereafter referred to as the *active constraint-weighted expected improvement* (ACW-EI) acquisition function. This acquisition function depends on both the GP models for CVaR and the expected return. In this paper, it is written as $a_{\text{CW-EI}}(\mathbf{w}, \widetilde{g}, \widetilde{R})$.

The new term $\text{PF}_{\max}$ in the acquisition function encourages the proposed points to be approximately active, which, by proxy, increases the likelihood that such a point is near-optimal with respect to the risk measure objective function. The choice of $r^{\max}$ is explored through additional numerical examples. Including this parameter is a crucial aspect of our proposed BO algorithms. Two feasible points with different true objective function values will likely have similar expected improvement values (before a full evaluation) due to the low-quality GP for the objective function and equal probability of feasibility for the constraint. As such, the two points may be considered equal in the existing methodology. By introducing the new $r^{\max}$ term - based on the more accurate expected return GP - our proposed BO procedure can differentiate between these two points during the selection procedure. Finally we note that there are certain existing acquisition functions may be of similar mathematical formulation as ACW-EI, e.g., Swersky et al. (2013); Gelbart et al. (2014); Wilson et al. (2018), but the purposes of them are very different from our ACW-EI acquisition function. In our algorithm ACW-EI is particularly designed for the constrained CVaR minimization problem, which provides the active constraint information to accelerate the computation, while most of the aforementioned works aim to design some general-purpose acquisition function with certain desired properties.

### 4.4 The Complete Algorithm

To complete our proposed algorithm, we must discuss the summation constraint:
$$0 \leq w_i \leq 1, \, i = 1, ..., N, \, \sum_{i=1}^{N} w_i \leq 1,$$
which will be denoted as $\mathbf{w} \in S$ in what follows. It is possible to deal with these constraints in the same manner as the expected return, i.e., as GP models. However, unlike the expected return constraint, which is probabilistic, the summation constraint is deterministic and easy to evaluate. As such, the constraint is imposed during the maximization of the acquisition function, by solving the following constrained maximization problem: $\max_{\mathbf{w} \in S} a_{\text{ACW-EI}}(\mathbf{w})$, which in this work is solved with the barrier method.

Finally, by combining the two-stage point selection, the ACW-EI acquisition function, and the constrained acquisition maximization, our complete *2S-ACW-EI BO* algorithm is obtained, detailed in Alg. 2.

### 4.5 Batch Implementation

In most BO approaches, one uses an acquisition function to select a single point to evaluate. From which, the posterior GPs are updated and the process is repeated. This is *sequential*, as each point is selected and evaluated one at a time.

---

**Algorithm 2** The 2S-ACW-EI BO algorithm

---
Initialize the training data sets $D$ (for the objective) and $C$ (for the constraint), using an initial design;
Let $t = 1$;
**while** stopping criteria not met **do**
    Construct a GP model $\widetilde{g}_{t-1}$ using $D$;
    Construct a GP model $\widetilde{R}_{t-1}$ using $C$;
    Let $\bar{\mathbf{w}} = \arg\max_{\mathbf{w} \in S} a_{\text{ACW-EI}}(\mathbf{w}, \widetilde{g}_{t-1}, \widetilde{R}_{t-1})$;
    Evaluate the constraint $R(\hat{\mathbf{w}})$;
    Let $C = C \cup \{\bar{\mathbf{w}}, R(\hat{\mathbf{w}})\}$;
    **if** $r^{\min} \le R(\hat{\mathbf{w}}) \le r^{\max}$ **then**
        Compute and estimate of the objective $g(\hat{\mathbf{w}})$, denoted as $\hat{g}$;
        Let $D = D \cup \{\hat{\mathbf{w}}, g(\hat{\mathbf{w}})\}$;
        let $t = t + 1$;
    **end if**
**end while**

---

It is expensive to evaluate the objective function, and as such, it may be advantageous to evaluate several points simultaneously, for example using parallel computers. In this regard, a batch implementation of BO is desirable, where several design points are selected using the acquisition function and then evaluated simultaneously in parallel. This section discusses a batch implementation for our proposed algorithms.

In most batch BO methods, the batch of design points is determined sequentially via a given point-selection procedure, from which the objective and constraint functions are evaluated after the whole batch is obtained. The batch implementation of the constraint-weighted expected improvement BO using the new acquisition function ('ACW-EI BO') is henceforth denoted *KB-ACW-EI BO*.

To adapt our two-stage BO algorithm for batch implementation, we include the evaluation of the expected return constraint in the point-selection procedure. Once the whole batch is obtained, the CVaR objective is evaluated in parallel. More specifically, the expected return is evaluated for each new proposed point. If the expected return satisfies Eq. 7, it is added to the batch and the constraint GP is updated. If the expected return does not satisfy Eq. 7, the point is not added to our batch, but the GP for the constraint is updated to ensure that the point is not proposed again. Once a batch has been determined, each point is fully evaluated - knowing that all batch points are both feasible and approximately active. The pseudo-code for our two-stage batch selection is provided in Alg. 3 - henceforth, denoted by *2S-KB-ACW-EI*.

The batch approach can be implemented in parallel, so it has a lower computational cost. However, the batch approach requires a greater total number of samples to converge to the optimal solution - as demonstrated in our numerical examples - due to the GPs being updated less frequently, so each sample is chosen based on a less accurate GP compared to at the equivalent stage in the sequential approach.

## 5 Numerical Experiments

We implement our proposed algorithms for two numerical examples and compare their results, and we also provided an additional application example in Appendix C. In all examples, BO was implemented using *Trieste* (Berkeley et al., 2023), a BO Python package built on TensorFlow. Within the package, we used the default Matern 52 Kernel, with length scale 1.0 and noise variance $10^{-7}$. For acquisition maximization, we include the summation constraint as a barrier function. The resulting problem is solved using the Efficient Global Optimization (EGO) method provided by the package.

### 5.1 Mathematical example

We first consider a simple mathematical example, to demonstrate how the design points are selected by the different methods. Adapted from Gramacy et al. (2016), we seek to solve the following constrained

---

**Algorithm 3** Two-Stage Batch Selection

---

**Require:** a training set for the CVaR objective function $D$, a training set for the expected return constraint $C$

**Ensure:** a batch of $b$ design points,

    let $B = \emptyset$

    let $i = 0$;

    **while** $i < b$ **do**

        propose a new design point $\bar{\mathbf{w}}$ based on a prescribed selection rule;

        evaluate the constraint $R(\bar{\mathbf{w}})$;

        **if** $r^{\min} \leq R(\bar{\mathbf{w}}) \leq r^{\max}$ **then**

            let $B = B \cup \{\bar{\mathbf{w}}\}$;

            let $i = i + 1$;

        **end if**

        let $C = C \cup \{\bar{\mathbf{w}}, R(\bar{\mathbf{w}})\}$;

        update the GP model for the constraint using $C$;

    **end while**

---

optimization problem:

$$\min_{\mathbf{w}} f(\mathbf{w}) := - w_1 - w_2$$
$$\text{s.t. } c(\mathbf{w}) := \frac{3}{2} - w_1 - 2w_2 - \frac{1}{2}\sin(2\pi(w_1^2 - 2w_2)) \geq 0 \tag{11}$$

The solution to the problem is $\mathbf{w} = (0.918, 0.540)$, where $f(\mathbf{w}) = 1.458$. The original CW-EI method, ACW-EI (i.e. the new acquisition function without the 2S process), and 2S-ACW-EI each use 10 initial points and then a further 50 iterations. Figure 1 shows the design points obtained by each of the three algorithms.

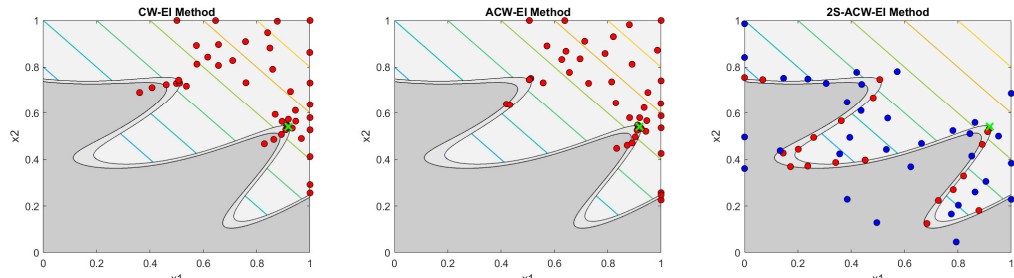

Figure 1: Plots showing the optimal solution (green-x) for numerical example one and the design points generated by each of the three methods. The figures include both the fully evaluated points (red) and those for which only the constraint was evaluated (blue). The feasible region is dark grey, the active region is light grey and the infeasible region is white. The objective function contours are shown too.

The CW-EI and ACW-EIs methods perform similarly in this task, where the algorithms generate a significant number of infeasible samples with high objective value, before moving towards the feasible region. Both methods first establish a good GP for the objective function, encouraging samples to be generated in the high objective region, before the GP for the constraint is fully formed. In contrast, in the 2S-ACW-EI method, samples are only fully evaluated if they are in the active region, therefore after a few iterations, the GP for the objective and constraint functions are weak and strong respectively. Thanks to the well-formed GP model for the constraint, the acquisition function prioritizes the generation of points in the feasible region, in particular, in the active region, before finding those feasible points which are maximized for the objective.

### 5.2 Portfolio allocation examples

### 5.2.1 Problem setup

The following three examples are based on an investor seeking to optimally allocate capital to stock or stock options, related to the twenty largest technology companies listed on American stock exchanges (both the NYSE & Nasdaq) by market capitalisation. We take $\mathbf{Z}$ to be the stock price at the future time, the distribution of which is determined by historical data. The parameter values are detailed in Appendix B.

In all three examples, the return function is

$$f(\mathbf{w}, \mathbf{z}) = \sum_{i=1}^{20} w_i y_i(z_i), \tag{12}$$

where $y_i$ is the asset return - stated as a ratio, rather than absolute value - corresponding to the $i$-th company, a function of its future stock price $z_i$. In the three examples, we alter the asset type – namely the function $y_i(z_i)$ varies. In each example, we consider a lower and higher return constraint.

**Example One.** We seek to optimally allocate the investor's capital directly to the twenty stocks, which corresponds to setting $y_i = z_i/\bar{z}_i$ with $\bar{z}_i$ being the stock's purchase price. Example 1 has a constraint for $1 - \alpha = 0.0001$ of $r^{min} =$ (a) 1.45 and (b) 1.55.

**Example Two.** We seek to allocate the investor's capital to European Call options, based on the twenty stocks, held till expiry. A European Call option gives the owner the right to purchase the underlying asset, for a pre-agreed strike price on a specified future date. Suppose that the present bid price of the call option for the $i$-th stock is $b_i$ and the strike price is $K_i$, then the asset return is
$$y_i = (\max(0, z_i - K_i) - b_i)/b_i$$
Example 2 has a constraint for $1 - \alpha = 0.0001$ of $r^{min} =$ (a) 5.30 and (b) 5.40.

**Example Three.** We consider European Call options, but where the return is derived from selling the option after six months rather than holding it to maturity. As such, the return depends on the change in the option price. Option prices can be modelled using quadratic functions of the underlying asset returns, realized through a delta-gamma approximation, that is, a second-order Taylor expansion of the portfolio return Zymler et al. (2013). Namely, at a particular future time, the associated call option return becomes

$$y_i = \Delta_i \, \epsilon + \frac{1}{2} \, \Gamma_i \, \epsilon^2,$$

with $\epsilon = z_i - \bar{z}_i$. Example 3 has a constraint for $1 - \alpha = 0.0001$ of $r^{min} =$ (a) 2.90 and (b) 3.00.

### 5.2.2 Experimental Results

In all three examples, we applied the three sequential methods and two batch methods: one based on standard CW-EI and also our proposed 2S batch method. We used 10 initial points, 110 iterations for the sequential methods and 11 batches of size 10 for the batch methods. In our numerical experiments, we set $r^{\max} = 110\% r^{\min}$ (i.e., $r^{\max}$ is 10% higher than the minimal expected return $r^{\min}$). All the experiments are repeated 20 times and the results are given in Table 1.

For all three examples, our proposed sequential methods outperform the standard BO approach, finding a lower CVaR objective value whilst meeting the feasibility condition. In addition, the two-stage approach produces better results than the one-stage approach. The same is true for the batch methods, where the two-stage method outperforms the one-stage method. The batch methods obtain better results than the standard sequential BO method but perform worse than the best sequential implementations. This is as expected, due to the GP only being updated after a full batch of samples has been identified, in contrast to the GP being updated after each new sample is proposed - as in the sequential approach. Using parallel implementation the batch method is significantly faster than the sequential approach. To further illustrate the results, we plot the best solution's objective value after each iteration in Figure 2. Consistently, the best solution of 2S-ACW-EI decreases faster than the other two sequential methods. The two-stage batch method performs better than the standard implementation, in all test cases.

| | Sequential BO Methods | | | Batch BO Methods | |
|---|---|---|---|---|---|
| | **CW-EI** | **ACW-EI** | **2S-ACW-EI** | **KB-ACW-EI** | **2S-KB-ACW-EI** |
| **1a CVaR (SD)** | 0.202 (0.013) | 0.199 (0.013) | **0.184** (0.012) | 0.199 (0.012) | 0.191 (0.012) |
| **1a Ex Return (SD)** | 1.473 (0.012) | 1.485 (0.012) | 1.473 (0.012) | 1.479 (0.012) | 1.478 (0.012) |
| **1b CVaR (SD)** | 0.266 (0.012) | 0.253 (0.012) | **0.247** (0.012) | 0.263 (0.014) | 0.249 (0.013) |
| **1b Ex Return (SD)** | 1.581 (0.012) | 1.577 (0.012) | 1.561 (0.012) | 1.580 (0.012) | 1.567 (0.012) |
| **2a CVaR (SD)** | 0.317 (0.013) | 0.291 (0.015) | **0.275** (0.014) | 0.302 (0.013) | 0.287 (0.013) |
| **2a Ex Return (SD)** | 5.335 (0.013) | 5.320 (0.012) | 5.302 (0.013) | 5.341 (0.012) | 5.322 (0.012) |
| **2b CVaR (SD)** | 0.336 (0.014) | 0.320 (0.014) | **0.303** (0.013) | 0.322 (0.013) | 0.308 (0.013) |
| **2b Ex Return (SD)** | 5.427 (0.013) | 5.428 (0.012) | 5.417 (0.013) | 5.433 (0.013) | 5.420 (0.012) |
| **3a CVaR (SD)** | −0.094 (0.012) | −0.122 (0.014) | **−0.132** (0.013) | −0.102 (0.012) | −0.131 (0.014) |
| **3a Ex Return (SD)** | 3.105 (0.013) | 3.030 (0.013) | 2.938 (0.012) | 3.082 (0.013) | 2.97 (0.013) |
| **3b CVaR (SD)** | −0.075 (0.013) | −0.083 (0.013) | **−0.094** (0.014) | −0.064 (0.012) | −0.075 (0.013) |
| **3b Ex Return (SD)** | 3.113 (0.013) | 3.075 (0.013) | 3.056 (0.012) | 3.125 (0.012) | 3.089 (0.013) |

Table 1: Average of the best objective and constraint values across repeated experiments: in each case, the best result among the methods is shown in bold. The standard deviations are given in parentheses.

| | Sequential BO Methods | | | Batch BO Methods | |
|---|---|---|---|---|---|
| | **CW-EI** | **ACW-EI** | **2S-ACW-EI** | **KB-ACW-EI** | **2S-KB-ACW-EI** |
| **1a CVaR (SD)** | 0.202 (0.013) | 0.198 (0.011) | **0.188** (0.014) | 0.201 (0.0.014) | 0.194 (0.011) |
| **1a Ex Return (SD)** | 1.473 (0.012) | 1.473 (0.018) | 1.471 (0.013) | 1.477 (0.016) | 1.474 (0.017) |
| **2a CVaR (SD)** | 0.317 (0.013) | 0.299 (0.014) | **0.281** (0.014) | 0.308 (0.011) | 0.293 (0.017) |
| **2a Ex Return (SD)** | 5.335 (0.013) | 5.324 (0.013) | 5.317 (0.016) | 5.331 (0.012) | 5.323 (0.013) |
| **3a CVaR (SD)** | −0.094 (0.012) | −0.115 (0.016) | **−0.125** (0.013) | −0.112 (0.014) | −0.128 (0.015) |
| **3a Ex Return (SD)** | 3.105 (0.013) | 3.103 (0.014) | 3.083 (0.018) | 3.102 (0.018) | 3.061 (0.013) |

Table 2: Same results as those in Table 1, but obtained with $r^{\mathrm{max}} = 105\% r^{\mathrm{min}}$.

Finally we want to note that a key parameter in the proposed algorithm is $r^{\mathrm{max}}$. Our numerical experiments found that setting $r^{\mathrm{max}} = 110\% r^{\mathrm{min}}$ generally works well. To test more rigorously how sensitive our proposed BO algorithm is to this parameter, we provide further numerical results obtained with $r^{\mathrm{max}} = 105\% r^{\mathrm{min}}$, in Table 2. These results are quantatively similar to those in Table 1 and thus show that our proposed algorithms are not highly sensitive to the choice of $r^{\mathrm{max}}$.

## 6 Conclusion

In summary, we consider the optimal portfolio allocation problem which aims to minimize a computationally demanding risk measure, subject to a minimum expected return constraint. We propose four new BO algorithms specifically designed for such problems that significantly reduce the number of evaluations of the expensive objective function. Furthermore, the proposed methods take advantage of the special properties of portfolio optimization problems, by developing a new acquisition function, a two-stage point selection process, and a batch implementation to take advantage of parallel computing. We expect that the proposed methods can be useful in problems arising from various fields, and in particular, we plan to explore its application to portfolio allocation problems in reinforcement learning (Ghosh et al., 2022) in the future.

Several issues of the proposed method should be addressed in the future. First of all the proposed method may not find the optimal solution for problems whose solutions do not lie on the boundary of the expected return constraint. In certain important real-world applications, such as automated investing, (financially) critical decisions are made upon the solutions of the optimal allocation problems. In this case returning a sub-optimal solution may have serious consequences, and as such it is highly desirable to impose mechanisms that can ensure the reliability and safety of the algorithms. For example, a heuristic strategy is to search around the obtained solution and see if a better one can be found. Due to its practical importance, this issue should be studied carefully in the future. Second a relevant class of methods are the contextual BOs Krause and Ong (2011); Letham and Bakshy (2019); Char et al. (2019); Feng et al. (2020), which aim to optimize the objective function with different contexts. Contextual BOs typically utilize a multi-GP model to take advantage of the correlation between different contextual awards. We here aim to optimize the single CVaR

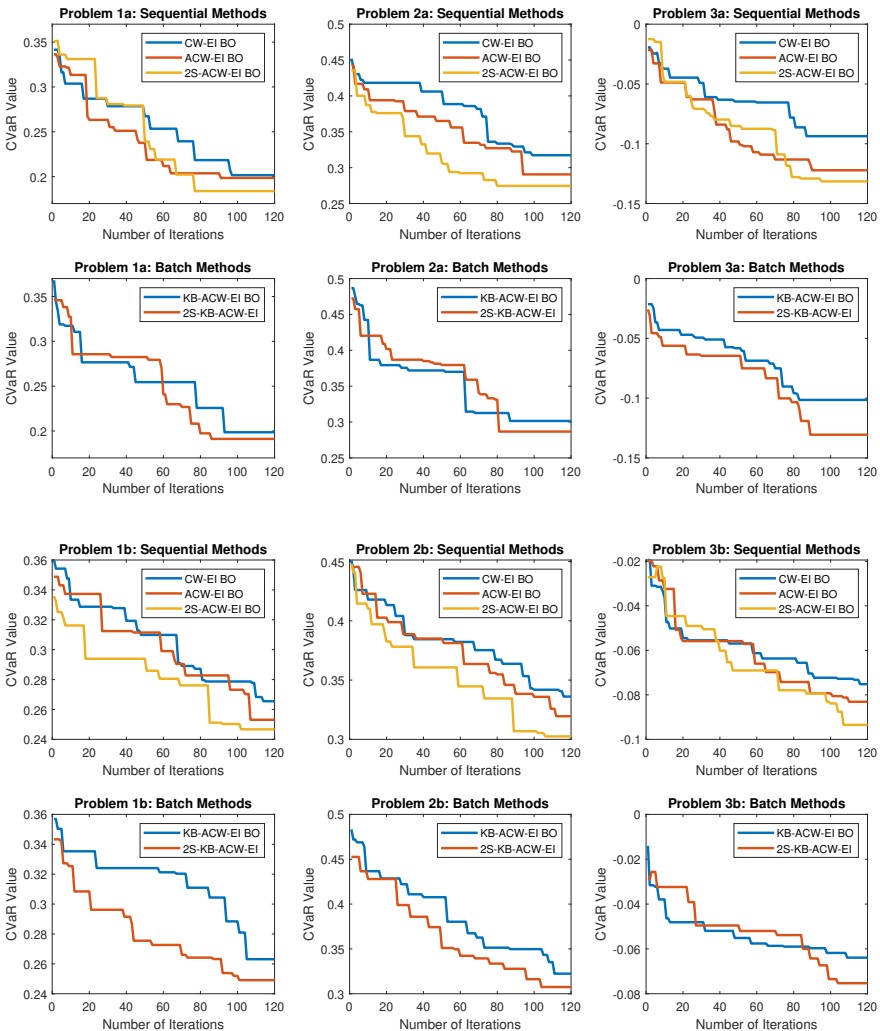

Figure 2: The best objective value obtained after each iteration for the portfolio allocation problems across the existing method (CW-EI BO) and the four new proposed methods.

function, and as such the contextual BO methods may not directly apply. That said, it is possible that the contextual BO can be extended to solve our problems, which is also a direction that we hope to explore.

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
