# OpenReview forum: "Bayesian Optimization for minimizing CVaR under performance constraints"
_TMLR — Rejected by TMLR_

### Review · Reviewer_85HT · 2024-03-07

**Summary Of Contributions:**

The paper proposes a Bayesian optimization method for portfolio allocation where the goal is to minimize risk as measured by CVaR under an expected performance constraint. The authors prove a theorem showing that under some technical conditions, an optimal allocation exists at the boundary of the feasible set. By exploiting this fact, they propose a two-stage procedure where only points near the boundary are really estimated for the risk measure. In addition, they propose an acquisition function to favor points near the boundary and consider a batch version of their algorithm to exploit parallel computation for the estimation of the risk measure.

**Audience:**

Yes

**Broader Impact Concerns:**

I believe this is not applicable for this work.

**Claims And Evidence:**

Yes

**Requested Changes:**

I believe that the authors should provide more discussion about the following points:
- Do the technical conditions of the theorem hold in the experiments?
- Could the technical conditions be relaxed?
- Can the proposed two-phase algorithm be suboptimal when the theorem does not hold?
- What happens if the proposed algorithm is run when Problem 1 has no solution, which can happen in practice if r_min is chosen too high?
- Have you run your algorithm with larger \alpha (e.g., 0.01 or 0.05)?
- Does PF(w) as defined in (8) really satisfy the first equality of (9)?

Other issues:
- In my opinion, the title is too generic. I think that it should be more precise and better reflect the content of the paper (e.g., mention CVaR minimization under performance constraint).
- The authors often use "i.e." instead of "e.g."
- References that are not part of a sentence should be between parentheses.
- Alg. 1 doesn't return a global minimizer but returns a solution in its neighborhood with high probability.
- Second to last sentence in paragraph above 4.1: I believe this sentence is meaningful only if you provide the alpha you have chosen.

Typos (just the first page):
resouces
suply chain
intellegence
Ghaoui et al. -> El Ghaoui

**Strengths And Weaknesses:**

STRENGTHS

The contributions are clearly outlined and well positioned with respect to existing work.

The algorithm seems overall reasonable.

The experiments demonstrate clearly the benefit of the two-phase method.

WEAKNESSES

While it's nice to have this theorem that justifies the two-phase method, it's not clear if it really holds in the experiments that are run and more generally how to make it hold in practice (e.g., choice of \alpha). I believe that the authors should discuss more those points and the consequences on the algorithm when their theorem does not hold.

The definition of the new acquisition function seems to be a bit heuristic, notably with first equation of (9).

The experiments are run with very small \alpha's, which are much smaller than what is used in practice in finance, I believe.

The paper should be prepared with more care. It contains quite a few typos and small issues with notations and phrasing.

---

> ### Author Response · Authors · 2024-03-26
> **Official Comment by Authors**
>
> Thanks a lot for your very constructive comments and we have made revision accordingly.
> First please be noted that the ``other issues'' identified in your review, including spelling, grammar and citation formatting have been corrected. The paper title has also been changed. Please see responses to your main questions below.
>
> 1. Do the technical conditions of the theorem hold in the experiments?
>
> We assume that the reviewer referred to Assumptions 1.  Assumptions 1 (a)-(c) are rather easy to verify and all the examples satisfy them. Assumption 1 (d), i.e. that on $\alpha$ is indeed  difficult to verify theoritically. We have verified 1 (d) numerically (i.e. via Monte Caro) in all the numerical examples.  An indirect evidence is that the results of the standard BO (CW-EI) agree with those of the proposed method, indicating that the solution is likely on the constraint boundary. That said, we do acknowledge that it is difficult to strictly ensure the requirement on alpha in advance. To this end, we also want to comment that we do not intend to use  the theorem to verify that an individual problem can be solved by the proposed algorithm. Rather, its main purpose is to demonstrate that it is not unrealistic to assume that many practical problems do admit the solution at the constraint boundary and therefore can be solved by the proposed method. We added some comments acknowledging the issue in 4.1 and the Conclusion section.
>
> 2. Could the technical conditions be relaxed?
>
> Of the four assumptions, 1(a) - 1(c) are hard to relax. It may be possible to relax assumption 1(d) to consider cases where no such $\alpha$ exists. Regarding relaxing 1(d) we also want to make the following two points. First, this is actually a "technical condition" that helps us to prove theorem 1. It may not be necessary, i.e., the theorem may hold under some alternative (possibly milder) conditions.  Second we would  also like to comment that, the scenariao that 1.(d) does not hold seems less interesting from a practical pointview.  Namely, in this case, there is no risk of loss but only risk of making less profit. Assessing and minizing such a risk seems less impactful, compared to the situation where it is possible that huge loss occurs.
>
> 3. Can the proposed two-phase algorithm be suboptimal when the theorem does not hold?
>
> It is possible -- since our method is motivated by the assumption that the optimal solution lies on the boundary, the proposed acquisition function strongly encourages the search to be focused on the constraint boundary. As such it is possible to settle with a suboptimal solution when the assumption does not hold. In the revised draft (Conclusion), we added discussion on the matter, indicating it is a main limitation of the proposed method. We also discuss some possible remedies for the issue.
>
> 4. What happens if the proposed algorithm is run when Problem 1 has no solution, which can happen in practice if $r_{min}$ is chosen too high?
>
> This is a general issue for the portfolio optimization problem. In the standard constrained BO (such as CW-EI), the new points will continuously be queried on (with both constraints and the objective function),  but will not be accepted (as it fails to satisfy the constraint). In our two-stage algorithm, the behavior is simlar except that the objective function is never evaluated at these points because they fail to satisfy the constraint.  In both algorithms, the iteration will be terminated after the maximum iteration number (specified in the stopping criteria) is reached.
>
> 5. Have you run your algorithm with larger $\alpha$ (e.g., 0.01 or 0.05)?
>
> Yes. The additional numerical example in the appendix sets $\alpha = 0.025$, which is the standard value used in industry. Under this example, the BO algorithms perform well compared to existing methods, across a longer time horizon. Within the second numerical example (in the main text), the alpha value is chosen to be very small to demonstrate that the algorithm can work under stricter conditions, with a greater computational cost.
>
> 6. Does PF(w) as defined in (8) really satisfy the first equality of (9)?
>
> Thanks a lot for pointing out the problem here. Upon checking the equations, we realised that one actually did not need Eq (9) and can evaluate $PF(w)$ directly from its definition. We revised the draft and removed Eq (9) completely.
>
> We are happy to answer any further questions and have more discussions in case some of your questions are not clearly answered.
>
> A revised draft has been uploaded.

---

### Review · Reviewer_ozr7 · 2024-03-11

**Summary Of Contributions:**

This paper proposes a Bayesian optimization policy for allotting financial portfolio.  In order to minimize a portfolio risk measure with particular constraints, the authors propose a new two-stage Bayesian optimization approach with a new acquisition function.  Finally, some experimental results are demonstrated to validate the proposed algorithm.

**Audience:**

Yes

**Broader Impact Concerns:**

I do not have any concerns on broader impacts.

**Claims And Evidence:**

No

**Requested Changes:**

- Please consistently use a sentence case or title case for the title of this work.  For example, the title should be "Optimal Portfolio Allocation Using Bayesian Optimization."
- In Section 1, `resouces` -> `resources`.
- Please use \citep and \citet correctly.
- In Section 1, `suply` -> `supply`.
- In Section 1, `artifical` -> `artificial`.
- The authors use `constraint function` frequently.  I think it should be `constrained function`.
- I think this problem is closely related to contextual Bayesian optimization.  It should be discussed and compared thoroughly.
- Why do the authors formulate Assumption 1 as a single assumption?  I think the authors can split it as four assumptions.
- In Theorem 1, `Assumptions 1` should be `Assumption 1`.
- In Theorem 1, `problem 1` should be `problem (1)`.
- Why is Theorem 1 required?  Is it really important?
- I think the form of the acquisition function proposed in this work is widely used in Bayesian optimization.  The relevant research should be cited appropriately.
- As described earlier, some contextual Bayesian optimization algorithms should be compared in the experiments.
- Could you draw standard deviations or standard errors in Figure 2?

**Strengths And Weaknesses:**

Strengths

- This work uses Bayesian optimization in interesting real-world applications.
- It provides thorough analyses on the proposed method.

Weaknesses

- Contributions are not clear.
- Some baselines are missing.
- Relationships to contextual Bayesian optimization should be discussed.
- Writing can be improved more.

---

> ### Author Response · Authors · 2024-03-31
> **Official Comment by Authors**
>
> Thanks a lot for your constructive comments. Issues relating to spelling, grammar and citation formatting have been corrected. Please see the responses to other comments. Our responses to the main requested changes are given below.
>
>
> 1.	The authors use constraint function frequently. I think it should be constrained function.
>
> Here we use “constraint function” to refer to the function that is a constraint, while we think “constrained function” means an objective function that is constrained, the meaning of which is different. As such we feel “constraint function” is more suited for our purpose. Happy to have more discussion.
>
>
> 2.	I think this problem is closely related to contextual Bayesian optimization. It should be discussed and compared thoroughly.
>
> Thanks for bringing the contextual BO (CBO) to our attention. We agree that our problem of interest shares certain similarity with the CBO methods. If we understand it correctly, the problems solved by CBO and our method are different. Namely, CBO intends to optimize a multi-objective optimization problem with one objective function per context, while in our setup, we have a standard single-objective function to optimize: CVaR(f(w,Z). As such it is sensible to use multi-GP for the former as they do have multiple objective functions. In our case, we have a single objective function that is the CVaR (it needs to be evaluated using a number of samples though), and so the multi GP based CBO approach is not directly applicable. In addition, we also note that the current single GP framework is not our contribution, it is similar to those used in e.g. (***). Our main contribution lies on using the constraint to accelerate the computation. To this end, even if  CBO can be modified and extend to our present problem, comparing them with the present single-GP formulation seem beyond the scope of our work. That said, we do agree that is very possible that CBO can be modified to solve the present problem which is interesting to study in the future. We added comments and references on CBO in the Conclusion section (blue texts). We are very happy to have more discussion and further clarification on the matter.
>
>
> 3.	Why do the authors formulate Assumption 1 as a single assumption? I think the authors can split it as four assumptions.
>
> This is a fair point. We have amended the paper to split the assumptions into four.
>
>
> 4.	Why is Theorem 1 required? Is it really important?
>
> Theorem 1 establishes the foundation of the proposed method. That is, under some practically sensible assumptions (i.e., Assumptions 1-4), the optimal solution lies on the boundary of the expected return constraint. This motivates our proposed algorithm that use this property to reduce the number of CVaR evaluations.  Without this property, in principle our method will not apply (or, as Reviewer 85HT stated, it will only find a sub-optimal solution).
>
>
> 5.	I think the form of the acquisition function proposed in this work is widely used in Bayesian optimization. The relevant research should be cited appropriately. As described earlier, some contextual Bayesian optimization algorithms should be compared in the experiments.
>
> We note that our acquisition function is ad hoc, i.e., it is designed to accelerate the type of portfolio optimization problem. It is not a general-propose acquisition function. As such, we believe that some existing acquisition functions may be of similar mathematical format as ours, but we have not found any of them are designed for similar purpose. We added a discussion and (some references) on the acquisition functions at the end of Section 4.3 (blue texts). It will also be very appreciated if you can let us know any specific references that you have in your mind but are overlooked by us. We provide some discussion regarding the contextual BO issue in Q2 and our understanding is that such methods are not directly applicable to our portfolio allocation problem. Please correct us if we are wrong.
>
>
> 6.	Could you draw standard deviations or standard errors in Figure 2?
>
> We think the reviewer for the sensible suggestion. A concern we have is that it may make the plots too busy and overshadow the comparison of the average performance. We also note that Table 1 provides such information. That said, we are happy to have more discussion on the matter and open to amend the figures in the next revision.
>
> We are happy to answer any further questions and have more discussions in case some of your questions are not clearly answered.
>
> A revised draft has been uploaded.

---

### Review · Reviewer_5eDa · 2024-03-13

**Summary Of Contributions:**

This paper proposes a new two-stage constrained Bayesian optimization method for the portfolio allocation problem. The first stage of the method filters out design points based on an acquisition function, and the second stage calculates the expected return, filters out points based on the expected return, and proceed to evaluate the objective function. These procedures encourage more efficiency and fewer evaluations on expensive objectives.

**Audience:**

Yes

**Broader Impact Concerns:**

I think this paper can benefit from having a statement. It is very relevant to automating investing / finance, so there might be some broader impact that can be clarified, e.g., what if the method fails, the assumptions are not satisfied, safety of investment etc.

**Claims And Evidence:**

No

**Requested Changes:**

1. Please clarify whether Z in section 2 is a random vector or a probability distribution. Both were mentioned. The authors seem to be mixing those two concepts. Similar for f(w, Z).

2. Bottom equation in page 2:
    - What is vf? This seems to be a typo.
    - What is the expectation over? It seems to me the right hand side should have lower case z, and the expectation is over $z\sim Z$, but the left hand side also has Z, so I'm fairly confused. This is also related to question 1.

3. Please add the definition of expected returns. Section 2.2 extensively talks about the relation between CVaR and expected returns, but it is unclear what the expected return is, and how it relates to all the symbols defined so far.

4. Explain what F and R are in Equation 1.

5. Explain $\tilde g$ in Algorithm 1. Typically in BO, the observations are g(x) with noise (as the authors mentioned). So, one cannot directly evaluate g(x), which is what the author wrote in the algorithm.

6. typo "Eq. equation 6" in page 4 under Eq 6.

7. In Section 2.1, it'd be nice to know what "subadditivity, translation invariance, positive homogeneity, and monotonicity" means and why VaR doesn't have those properties.

8. Clarify if $f$ at the bottom of page 4 is still the $f$ in page 1, and what $x$ (point to optimize) corresponds to for function $f$.

9. To make the literature review more compact, Section 2 and 3 can potentially be combined into the same section called literature review or something like that. The notations can be unified, and the authors can talk about Bayesian optimization / constrained BO in the context of notations in Section 2.

10. Clarify what $r^{min}$ exactly is, similarly for $r^{max}$. If $r^{min}$ is arbitrarily set, I don't see how Theorem can hold. If it is set to be

11. It is unclear what the function to be optimized really is in this work. Is it f (in Eq 11)? Is it F (in Eq 1a)? Or is it something else? It'd be nice to clarify it early on, especially when introducing the $g$ function in the BO section.

**Strengths And Weaknesses:**

Strengths:
- Portfolio optimization is a very interesting application of BO and I think it has a lot of potential.
- I like that the authors have sections on background in Section 2/3. They are helpful for people to understand the literature. Though I think it'd be nice to make it clearer in terms of what is new and what is a literature review/background. I'm also slightly concerned that Section 2/3 occupy a lot of space of this paper. Some efforts can be made to make they more compact. See Requested Changes.

Weaknesses:
- Clarity can be improved. See Requested Changes. The paper became difficult to read without clear definition of several critical terms, such as  $r^{min}$, expected return, R, F etc.
- The method builds on constrained BO work in 2014, but there has been new advancement on constrained BO. It is not entirely clear why the authors chose the work in 2014 to build upon.
- The adaptation of constrained BO in this work is somewhat incremental from a machine learning perspective. This work might be more suited for the finance / operations research community.
- It is not very clear to me how expensive the objective evaluations really are for the application the authors studied. The example in Eq 12 seems like a linear function over numerical values that can be easily computed.
- Baselines are limited. It might be good to add other constrained BO methods, like those mentioned in Section 1 .
- Overall, it is not very clear how useful the BO part of this method is for portfolio allocation since BO seems to be the more expensive part, with all the GP updates etc.

---

> ### Author Response · Authors · 2024-03-29
> **Official Comment by Authors**
>
> Thanks a lot for your constructive comments. Issues relating to spelling, grammar and citation formatting have been corrected. Please see the responses to other comments. Our responses to the specific requested changes are given below.
>
> 1. Please clarify whether Z in section 2 is a random vector or a probability distribution...
>
> Both $Z$ and $z$ are vectors representing the same physical parameters practically.
> When we are referring them as a random variable we use $Z$, e.g., $E_Z[f(w,Z)$,  and when it standards for a specific realisation (or a specific value) we use $z$, e.g., $\sum_{i=1}^nf(w,z_i)/n$. As you pointed out later, they were used mistakenly in some equations which have been corrected in the revised draft. We clarify this in the 2nd paragraph of Section 2.
>
> 2. Bottom equation in page 2: (a) What is vf? This seems to be a typo. (b) What is the expectation over? ... This is also related to question 1.
>
> (a) We have amended the definitions of VaR and CVaR to make this clearer.
> (b) You are correct. We have amended the definition of the expectation return to clarify this.
>
> 3. Please add the definition of expected returns. Section 2.2 extensively talks about the relation between CVaR and expected returns, but it is unclear what the expected return is, and how it relates to all the symbols defined so far.
>
> We have included the definition of expected return at the beginning of section 2.2.
>
> 4. Explain what F and R are in Equation 1.
>
> We have revised the notations following your suggestion (9) and we hope that the new notations resolve the problem. We define R as the expected return $R(w)=E_Z[f(w,Z)]$ in Eq (1b).
>
> 5. Explain $g^{~}$ in Algorithm 1. Typically in BO, the observations are g(x) with noise (as the authors mentioned)...
>
> We agree and revised the draft accordingly. Now we assume that a noisy estimate of $g(x)$ can be obtained. Please see Alg. 1 and Section 3.2 (texts in blue)
>
> 6. Typo "Eq. equation 6" in page 4 under Eq 6.
>
> Corrected.
>
> 7. In Section 2.1, it'd be nice to know what "subadditivity, translation invariance, positive homogeneity, and monotonicity" means and why VaR doesn't have those properties.
> Agreed and a footnote has been added in Section 2.1 to include these ley definitions.
>
> 8. Clarify if f at the bottom of page 4 is still the f in page 1, and what x (point to optimize) corresponds to for function f.
>
> They are the same. We are trying to explain the strategy used in the aforementioned references -- model the underlying function f(w,Z) rather than the risk measure as GP. The design parameter is now $w$. Again we revised the notations throughout the paper, and we hope they are more clear now.
>
> 9. ..., Section 2 and 3 can potentially be combined into the same section ... The notations can be unified...
>
> We agree completely that the notations should be unified, and we revised the draft accordingly. In particular the design parameter (parameter to optimize) is $w$ and the objective function is $g(w)$ throughout the paper. Regarding your suggestion to combine Sections 2&3, while we appreciate the advantages of doing so, we slightly prefer to have two separate sections with one on the portfolio optimization problem and one on BO algorithms. An alternative is to have a “background and preliminaries” section with two subsections (one on portfolio optimization and one on BO). We are open to further discussion on the matter.
>
> 10. Clarify what $r^{min}$ exactly is, similarly for $r^{max}$. If $r^{min}$ is arbitrarily set, I don't see how Theorem can hold...
>
> $r^{min}$ is defined within Section 2.2, as the minimum expected return an investor seeks (i.e., it is not an algorithm parameter and is set by the users). The point you raise is fair, in that the return an investor seeks must be within reason of the proposed assets and obtainable from the probability distribution of return forecasts. This assumptions are widely accepted within the literature and application context.
>
> 11. It is unclear what the function to be optimized really is in this work. Is it f (in Eq 11)? Is it F (in Eq 1a)? Or is it something else? It'd be nice to clarify it early on, especially when introducing the g function in the BO section.
>
> Following your suggestions above we revised the notations and so now the are unified. We now use  $g(w)$ to represent the objective function: it is defined in Eq (1.a) which also demonstrates how g(w) is related to f(w,Z). To further address this issue, we have added an additional paragraph to the end of section 2.2 (texts in blue).
>
> 12. It is very relevant to automating investing / finance, so there might be some broader impact that can be clarified, e.g., what if the method fails...
>
> It is a very useful suggestion and we added such a discussion in the Conclusion section (blue texts).
>
> We are happy to answer any further questions and have more discussions in case some of your questions are not clearly answered.
>
> A revised draft has been uploaded.

---

### Decision · Action_Editor_1vAd · 2024-05-09

**Recommendation:** Reject

**Comment:**

In addition to the technical questions/issues, the reviewers were concerned with the overall quality of the submission with quite a few typos, and issues with notations and phrasing. Although the authors corrected some of them during the rebuttal, the manuscript should be prepared with more care and all these issues need to be fixed before submission.

In addition to the presentation issues, the reviewers still have several technical concerns. I listed some under "claims and evidence". Others include
1) The method builds on a relatively old (2014) constrained BO work, while there has been new advancement on constrained BO. The reasoning behind this choice is not quite clear.
2) The work is somewhat incremental and can be seen more as an application in finance or OR, in which case more careful evaluation is required.
3) The cost of BO needs to be discussed in more details to justify its usefulness for this particular application.

Overall, none of the reviewers thinks the paper is ready for publication in its current form.

**Audience:**

The findings of the paper could be of interest to some TMLR's audience.

**Claims And Evidence:**

In this paper, the authors propose a Bayesian optimization algorithm for portfolio allocation. The goal is to minimize risk, measured by CVaR, while satisfying a constraint on expected performance. They prove that under several technical assumptions, an optimal allocation can be found at the boundary of the feasible set. Using this finding, they propose a two-stage algorithm in which in the first stage the design points are filtered out by an acquisition function that favors points near the boundary, and in the second stage it filters out points based on the expected return and then evaluates the objective function. They consider the batch version of their algorithm to exploit parallel computation for estimating the risk measure.

The reviewers have the following concerns about the results/claims:
1) It is not clear if the assumptions used in the main theorem (e.g., choice of \alpha) hold in the experiments or how to make sure that they hold in practice. Unfortunately, the authors do not discuss this and its practical consequences.
2) It seems the values of \alpha used in the experiments are much smaller than those in practice (in portfolio allocation).
3) Not enough baselines have been used in the experiments. There are other constrained BO methods that can be used in the experiments.

**Resubmission Of Major Revision:**

The authors may consider submitting a major revision at a later time.